# Divide, conquer and reconstruct: How to solve the 3D structure of recalcitrant Micro-Exon Gene (MEG) protein from *Schistosoma mansoni*

**Stepanka Nedvedova**[1,2,3], **Florence Guillière**[1], **Adriana Erica Miele**[1,4], **François-Xavier Cantrelle**[5], **Jan Dvorak**[3,6,7], **Olivier Walker**[1], **Maggy Hologne**[1] *

1 Université de Lyon, CNRS, UCB Lyon1, Institut des Sciences Analytiques, UMR5280, 5 rue de la Doua, Villeurbanne, France, 2 Department of Chemistry, Faculty of Agrobiology, Food and Natural Resources, Czech University of Life Sciences Prague, Prague, Czech Republic, 3 Department of Zoology and Fisheries, Center of Infectious Animal Diseases, Czech University of Life Sciences Prague, Prague, Czech Republic, 4 Department of Biochemical Sciences, Sapienza University of Rome, Rome, Italy, 5 Université de Lille, CNRS, UMR8576 –UGSF–Unité de Glycobiologie Structurale et Fonctionnelle, Lille, France, 6 Institute of Organic Chemistry and Biochemistry, Czech Academy of Sciences, Prague, Czechia, 7 Faculty of Environmental Sciences, Center of Infectious Animal Diseases, Czech University of Life Sciences in Prague, Prague, Czech Republic

* maggy.hologne@univ-lyon1.fr

**Data Availability Statement:** NMR assignment data have been deposited into the BMRB database under the following accession numbers: 51844

## Abstract

Micro-Exon Genes are a widespread class of genes known for their high variability, widespread in the genome of parasitic trematodes such as *Schistosoma mansoni*. In this study, we present a strategy that allowed us to solve the structures of three alternatively spliced isoforms from the *Schistoma mansoni* MEG 2.1 family for the first time. All isoforms are hydrophobic, intrinsically disordered, and recalcitrant to be expressed in high yield in heterologous hosts. We resorted to the chemical synthesis of shorter pieces, before reconstructing the entire sequence. Here, we show that isoform 1 partially folds in a-helix in the presence of trifluoroethanol while isoform 2 features two rigid elbows, that maintain the peptide as disordered, preventing any structuring. Finally, isoform 3 is dominated by the signal peptide, which folds into a-helix. We demonstrated that combining biophysical techniques, like circular dichroism and nuclear magnetic resonance at natural abundance, with *in silico* molecular dynamics simulation for isoform 1 only, was the key to solve the structure of MEG 2.1. Our results provide a crucial piece to the puzzle of this elusive and highly variable class of proteins.

## Introduction

Schistosomes are digenean parasitic worms with a complex life cycle and perfectly adapted to both intermediate (mollusk) and definitive host (mammal) [1, 2]. In humans, the chronic disease is caused by the inflammatory reactions elicited by the eggs trapped in the host tissues,

(isoform 1a), 51837 (isoform 1b), 51846 (isoform 1c), 51845 (isoform 1d), 51852 (isoform 2a), 51851 (isoform 2b) and 51853 (isoform 3).

**Funding:** This research was funded by Improvement in Quality of the Internal Grant Scheme at CZU, reg. no. CZ.02.2.69/0.0/0.0/19_073/0016944, financed from the funds of Operational Programme Research, Development, and Education, in the framework of ESF Call no. 02_19_073 for Improving the Quality of Internal Grant Schemes at Higher Educational Institutions in priority axis 2 OP. The funders had no role in study design, data collection and analysis, decision to publish, or preparation of the manuscript.

**Competing interests:** The authors have declared that no competing interests exist.

**Abbreviations:** MEG, Mico-Exon Genes; IDP, Intrinsically Disordered Protein; CD, Circular Dichroism; NMR, Nuclear Magnetic Resonance; DMSO, dimethylsulfoxide; TFE, trifluoroethanol; HSQC, Heteronuclear Single Quantum Correlation; NOESY, Nuclear Overhauser Spectroscopy; TOCSY, Total Correlation Spectroscopy; MD, Molecular Dynamics; RMSD, Root Mean Square Deviation; RMSF, Root Mean Square Fluctuations.

while adult worms can live undisturbed for years in the blood flow [1, 3]. Many molecular studies flourished after completing of *Schistosoma mansoni* genome sequencing [4, 5],. The analysis of secreted biomolecules by eggs, adult and juvenile worms highlighted the abundance of a peculiar family of highly variable peptides/proteins, ranging from 7 to 20 kDa, without homologs in the host, called Micro-Exon Gene (MEG) proteins [6–8]. The genetic structure of micro-exons is not unique to schistosomes, but proteins annotated in the UniProt database [9] (Release 2022_05) as schistosome MEG proteins, as much as 109, do not show relevant homology to other proteins in this database. The name of the protein family is derived from its distinctive genetic characteristic: up to 75% of the gene is composed of 3 to 36 base pairs exons, which can undergo alternative splicing, resulting in an enormous protein variability [4, 8].

In the *Schistosoma mansoni* life cycle, MEGs are preferentially expressed in the intra-mammalian life cycle stages. Previous studies have identified two super-families of MEGs, MEG 2 and MEG 3, as being particularly abundant in egg secretions [5, 6, 8].

Therefore, we focused on three isoforms of the MEG 2.1 family, namely MEG 2.1 full-length isoform 1 (UniProt ID D7PD78) and its two alternatively spliced isoforms MEG 2.1 isoform 2 (UniProt ID D7PD76) and MEG 2.1 isoform 3 (UniProt ID D7PD75). In the present article, we present for the first time the 3D structure of members of the MEG family obtained using Nuclear Magnetic Resonance (NMR) experiments at $^{15}$N and $^{13}$C natural abundances and circular dichroism (CD) experiments. To achieve our goal, isoform 1 as well as isoform 2 were divided into shorter peptides and a divide, conquer and reconstruct strategy was carried out to get structural information. All the peptides have an intrinsically disorder nature (IDP), however, after performing MD simulation on reconstructed isoform 1, we highlighted the formation of a-helix at the N-terminus of isoform 1 as deduced by CD analysis in the presence of TFE. This result points to the possible morphing from IDP to helical structure induced by the local amino acid sequence or by a binding partner.

## Materials and methods

### MEG 2.1 protein production

MEG 2.1 isoform 1 (UniProt ID: D7PD78) from *S. mansoni* was cloned into pET22b and pET-Sumo expression vectors with a 6xHis tag to help in the purification step (Figure S1 in S1 File). These plasmids were then transformed into the following commercial strains of *Escherichia coli*: BL21(DE3)Gold, One shot BL21(DE3)pLysS, Rosetta (DE3), Rosetta (DE3)pLysS. Despite testing different expression conditions, such as temperature, medium, induction time, and inductor concentration, the protein was found to be either toxic to the bacteria, as shown in Figures S1 and S2 (S1 File), or expressed in meagre yield. We also tried to express the proteins in methanotrophic yeast *Komagatella phaffii* (formerly known as *Pichia pastoris*) as well as using *in vitro* (cell-free) expression system (Figures S3 and S4 in S1 File), but again the yield was too low to proceed to structural studies (see the full description of the experimental conditions and results in the S1 File).

### Chemically synthesized peptides

Since no expression protocol was successful, we decided to break the sequences of the two isoforms of *S. mansoni* MEG 2.1 (UniProt ID D7PD78 and D7PD76) into shorter peptides and chemically synthesized them as shown in Fig 1. All the peptides used for the analyses were chemically synthesized by Genosphere Biotechnologies (https://www.genosphere-biotech.com). Isoform 3 (1–26) and isoform 1 (25–88) as well as all the shortest four peptides iso 1a (25–43), iso 1b (42–58), iso 1c (58–72) and iso 1d (73–88) were synthesized to a minimum of 95% purity in natural abundance. After three attempts, the company was unable to synthesize

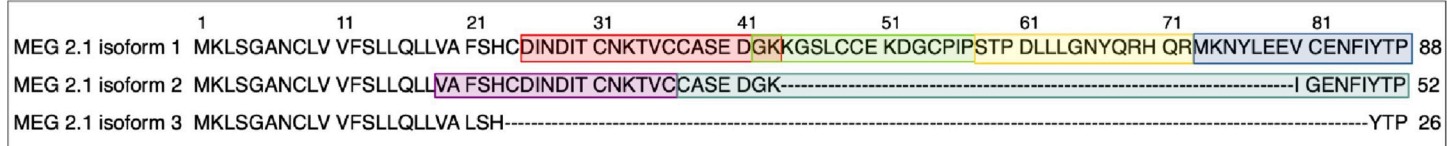

**Fig 1. Sequence alignment of *S. mansoni* MEG 2.1 proteins.** MEG 2.1 isoform 1 corresponds to UniProt ID D7PD78. The synthesized peptide sequences of iso 1a, iso 1b, iso 1c and iso 1d are displayed in coral, green, yellow, and blue respectively. MEG 2.1 isoform 2 corresponds to UniProt ID D7PD76. The synthesized peptide sequences of iso 2a and iso 2b are displayed in purple and petrol green respectively. MEG 2.1 isoform 3 corresponds to UniProt ID D7PD75.

isoform 2 (19–52). Hence, we decided to split the sequence into two short peptides iso 2a (19–36) and iso 2b (37–52), which were successfully chemically synthesized by Genosphere. The peptides were hardly soluble in biological buffers such as Tris/HCl, phosphate or MES buffer, independently of the salt concentration. We also unsuccessfully tried organic solvents, such as chloroform, acetone, methanol, and isopropanol. To be able to dissolve them at 2 mM we resorted to DMSO.

## Circular dichroism (CD) experiments

CD spectra were acquired for the three chemically synthesized MEG 2.1 isoforms–isoform 1, isoform 2 (iso 2a, iso 2b) and isoform 3 (iso 3) peptides on a Chirascan VX spectrometer (Applied Photophysics). Since DMSO is unsuitable for CD analyses, we used acetonitrile, where the peptides were partially soluble. Acetonitrile proved to be a suitable solvent for CD measurements as it does not absorb between 180 and 250 nm, where the features essential for protein/peptide secondary structure lie. After solubilization and subsequent centrifugation, the samples were prepared from the resuspended peptide in acetonitrile supernatant. The initial peptide concentration was set to 10 μM. The samples were measured at 20°C in a HELLMA Macro-Cuvette 100-QS 1mm Quartz Glass 100-1-40 in the range from 180 nm to 280 nm with a step size of 0.5 nm, a bandwidth of 1 nm and in five repetitions for the entire spectrum. The spectra presented in Fig 2 are the average of the 5 measures after subtracting the average of 5 baselines.

## NMR chemical shifts assignment of the peptides

NMR experiments were carried out at 27°C using a Varian Inova spectrometer operated at a $^1$H frequency of 600 MHz (14.1 T) and equipped with a triple HCN cryoprobe enhanced in $^1$H and $^{13}$C. Each peptide was prepared by dissolving the synthetic peptide in 450 μL DMSO-d6 (Eurisotop) to a concentration of 2 mM. For proton resonance assignments we used homonuclear experiments as zero quantum TOCSY (zTOCSY) experiment [10] with 16 scans and mixing time of 80 ms and 100 ms; NOESY experiment [10] with 28 scans and a mixing time of 400 ms. For $^{15}$N and $^{13}$C resonance assignments we used heteronuclear experiments in natural abundance: $^1$H-$^{15}$N HSQC experiment [11] with 600 scans and $^1$H-$^{13}$C HSQC experiment with 256 scans, with and without $^1$H-$^{13}$C multiplicity editing. To complete $^1$H and $^{13}$C assignments we also acquired one $^1$H-$^{13}$C HSQC-TOCSY experiment (adapted from [12, 13]) with 400 scans and a mixing time of 80 ms. Except for the $^1$H-$^{15}$N HSQC experiment, we have used a double pre-saturation for each 2D experiment at 2.48 ppm and 3.33 ppm to suppress signals of DMSO and $H_2O$, respectively. To monitor the peptide stability, between each 2D experiment, we have inserted a standard 1D $^1$H experiment with the same double pre-saturation for DMSO and $H_2O$ signals.

In the case of uncut isoform 1 (25–88) and isoform 3, the NMR experiments were also carried out at 27°C with a Bruker NEO spectrometer operated at a $^1$H frequency of 1.2 GHz (28.2

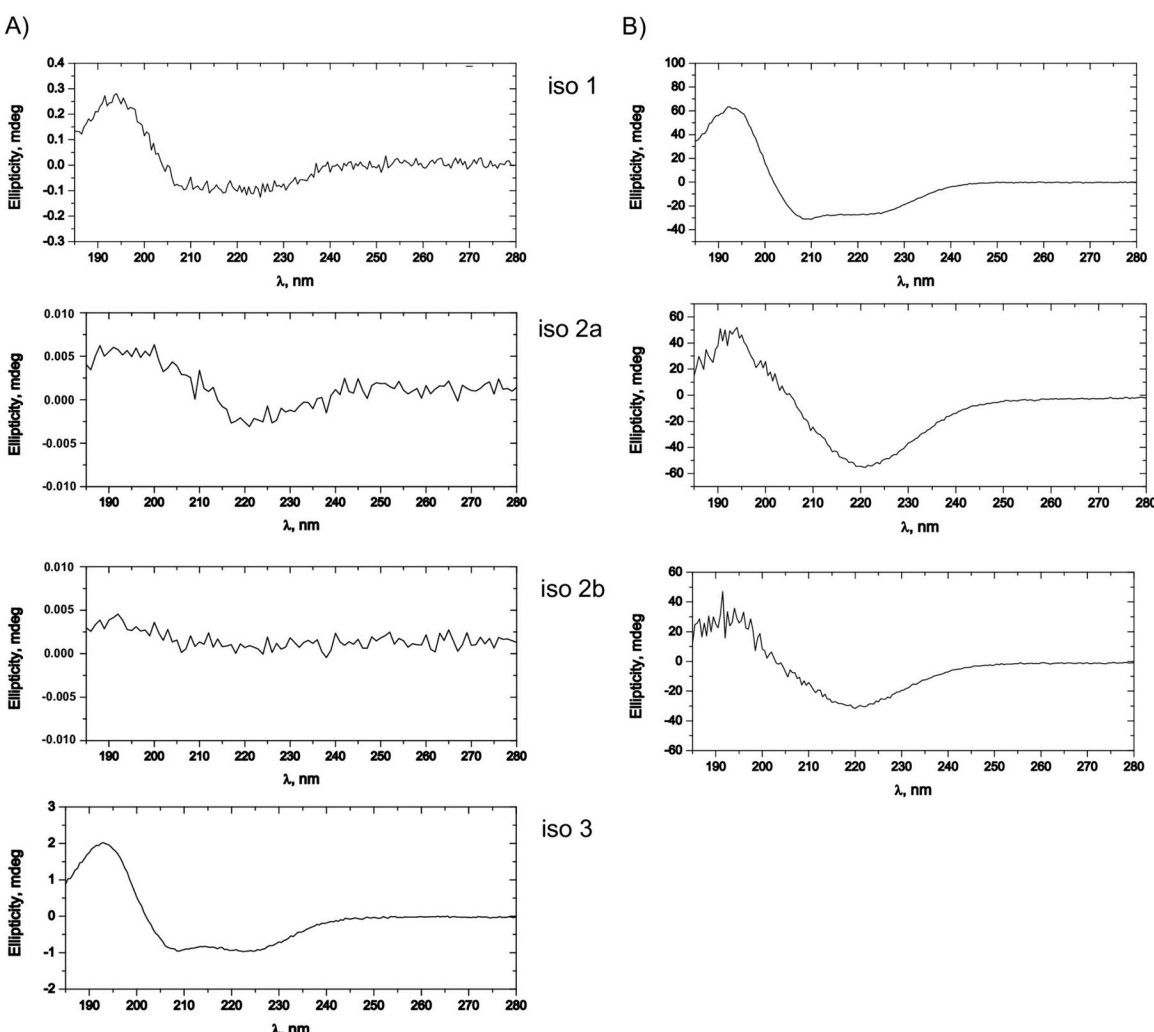

**Fig 2. CD spectra of MEG 2.1 isoforms.** The CD spectra of chemically synthesized isoform 1, isoform 2a, isoform 2b and isoform 3 were recorded at 25°C in 100% acetonitrile (Panel A) and 50% acetonitrile + 50% TFE (Panel B). Spectra were recorded for the supernatant after centrifugation. The initial peptide concentration was set to 10 μM.

T) and equipped with a triple HCN cryoprobe. Each sample was prepared by dissolving the peptide in 200 μL DMSO-d6 (Eurisotop) to a concentration of 2 mM. The following experiments were recorded: $^1$H-$^1$H TOCSY [14] (mixing time = 60 ms, 16 scans), $^1$H-$^1$H NOESY (mixing time = 120 ms, 48 scans), $^1$H-$^{13}$C edited HSQC [15] (8 scans) and sofast $^1$H-$^{15}$N HSQC [16] (256 scans).

2D data were processed using nmrPipe and nmrDraw [17] processing software. Peak assignment and visualization were performed using Sparky [18] and Poky [19] softwares.

## 3D structure refinement of short peptides

CYANA [20] version 2.1 was used for automatic structure calculations based on assigned spectra ($^1$H-$^1$H NOESY, $^1$H-$^{13}$C HSQC and $^1$H-$^{15}$N HSQC). The algorithm used by the CYANA program is based on a consistent probabilistic treatment of the NOE assignment process. The goal of the procedure is to reduce ambiguities of NOE assignments that could lead to erroneous distance restraints. The structure calculation is an iterative process over 7 cycles. The same

input data are used for all cycles and comprise the assigned chemical shift list, the amino acid sequence of the peptides and a list containing the positions and the volume of the NOE cross peaks measured in the 2D $^1$H-$^1$H NOESY spectrum. Finally, the unambiguous distance restraints are included as input of the structure calculation with simulated annealing by the fast CYANA algorithm developed by Güntert et al. [21]. The precision of the calculated structures improves with each subsequent cycle.

## Reconstruction of isoform 1 (25–88) structure

The NMR peptide structures (iso 1a, iso 1b, iso 1c and iso 1d) obtained after CYANA treatment were built together with UCSF Chimera software using the default setting Join Models section for C-N peptide bond [22]. The 4 amino acids FSHC were also added to the N-terminal part of the reconstructed peptide to design the complete sequence without the signal peptide (corresponding to residues 1 to 20). Such reconstructed models were subjected to energy minimization using the Chiron protein structure refinement server [23] that minimizes steric clashes in proteins using short discrete molecular dynamics (DMD) simulations until it attains an acceptable clash score. The resulting protein structure has normalized clash score comparable to high-resolution protein structures ($< 2.5$ Å).

## Molecular dynamics (MD) simulation

MD simulations were carried out using the ACEMD3 software [24] with the Amberff14SB force field [25] and TIP3P water model [26]. The system was minimized and equilibrated under constant pressure (1 atm) and temperature (300 K) conditions using a time step of 4 fs, a non-bonded cutoff of 9 Å, and particle-mesh Ewald long-range electrostatics with a grid of $52 \times 81 \times 103$ with a spacing of 1 Å. The system was first equilibrated during 10 ns, then a production run of 4 μs was performed. The processing of the trajectory and the secondary structure timeline analyses were processed by using VMD [27] and the GROMACS dssp utilities [28, 29]. The radius of gyration (Rg), the Root Mean Square Deviation (RMSD), and the Root Mean Square Fluctuations (RMSF) parameters were calculated by means of the corresponding GROMACS tools.

## Results

### Strategy for MEG 2.1 isoforms 1, 2, and 3 productions

The sequence of isoforms 1, 2 and 3 of *S. mansoni* MEG 2.1 are displayed in Fig 1. To circumvent the low yield in production, the isoform 1 (25–88) and 3 (1–26) were chemically synthesized to a minimum of 95% purity in natural abundance by Genosphere Biotechnologies (https://www.genosphere-biotech.com). According to the SignalP database, residues 1–20 of isoforms 1 and 2 are predicted as a putative signal peptide and show a practically complete identity with isoform 3. Unfortunately, the company failed after three trials to synthesize isoform 2 (19–52). Consequently, we decided to break the sequence of isoform 2 into two shorter chemically synthesized peptides (Fig 1): iso 2a (residues V19 to C36) and iso 2b (residues C37 to P52). To prevent any bottleneck in NMR analyses due to crowded signals of the 64 residues, isoform 1 was also divided into four shorter peptides, namely iso 1a (residues D25 to K43), iso 1b (residues G42 to S58), iso 1c (residues S58 to R72) and iso 1d (residues M73 to P88). As an example, this "divide and conquer" strategy was previously successfully used by Chabert et al. to solve structures of peptides in natural abundance issued from the protein SilE [30]. Another challenge faced with the synthetic peptides was solubilizing them in an appropriate solvent

suitable to all the structural analyses. According to the primary structure analysis the peptides are predicted to be poorly soluble (Table 1), which was indeed the case.

First, we tried the aqueous buffers commonly used for peptides or proteins such as phosphate, MES or Tris-HCl buffers (20 mM) at pH ranging from 6.5 to 8. Unfortunately, the solubility was dramatically poor despite adding NaCl (150 mM) or b-mercaptoethanol (up to 500 μM). Pure organic solvents such as acetone, chloroform or trifluoroethanol (TFE), were tested unsuccessfully. Moreover, the challenge was finding a suitable solvent for both CD and NMR experiments according to the different solvents' peptide concentrations and optical activity. Indeed, CD experiments can be carried out at low peptide concentrations (10 μM), but the solvents such as methanol, isopropanol or DMSO were unsuitable. Finally, we found that acetonitrile was able to solubilize the peptides at 5–10 μM and had no significant CD absorbance. However, in order to perform NMR experiments on peptides containing $^{15}$N and $^{13}$C in natural abundance (respectively 1.1% and 0.4%), the minimal concentration was set to 2 mM. Only DMSO allowed to dissolve the peptides at this concentration completely.

## Secondary structure analysis by circular dichroism (CD)

We resuspended the peptides in acetonitrile for 2D structural analysis by circular dichroism (CD). The quantity of resuspended peptide was extremely poor, and abundant precipitation was present, as outlined in the previous section. Acetonitrile is not known to be used for peptide structure determination but Bocian et al. showed that this it was suitable for studying the native conformation of human insulin monomer [32, 33] and could be used in case of structure elucidation in water is difficult due to precipitation or aggregation. Zhu et al. also solved the structure of a-momorcharin dissolved in 80% acetonitrile [34].

Despite these challenges, a series of CD spectra were recorded, as displayed in Fig 2. The spectra were measured in both the absence and presence of 50% TFE. For the two peptides of isoform 2, the CD spectra are relatively noisy and present a single negative peak at around 220 nm, the hallmark of disordered protein (Fig 2A). The same feature was also observed in 1:1 acetonitrile:TFE, indicating no induction of a-helices in these two peptides (Fig 2B), whereas TFE is known to enhance helicity in polypeptides [35, 36]. On the other hand, the CD spectra of MEG 2.1 isoform 1 (residues 25–88) and isoform 3 showed a positive ellipticity value at 190 nm and negative ellipticity values at 208 and 222 nm, indicating the presence of a-helical regions, a feature enhanced in the presence of 50% TFE. Moreover, in all cases TFE improved the solubilization of the peptides.

**Table 1. GRAVY index.** Summary of hydropathicity index (GRAVY [31]) for the MEG 2.1 peptides in this study. An increasing positive score indicates greater hydrophobicity. Glycine index is -0.4.

| MEG 2.1 peptide (residues number) | GRAVY index |
|---|---|
| Isoform 1 (25–88) | -0.608 |
| Iso 1a (25–43) | -0.468 |
| Iso 1b (42–58) | -0.524 |
| Iso 1c (58–72) | -1.307 |
| Iso 1d (73–88) | -0.653 |
| Iso 2a (19–36) | +0.344 |
| Iso 2b (37–52) | -0.438 |
| Isoform 3 (1–26) | +1.136 |

## NMR chemical shifts assignment of peptides

We have recorded the following 1D and 2D NMR experiments for each chemically synthesized peptide (2 mM) dissolved in deuterated dimethyl sulfoxide (DMSO-d6) at natural abundance: 1D $^1$H, 2D $^1$H-$^1$H TOCSY, 2D $^1$H-$^1$H NOESY, 2D $^1$H-$^{13}$C HSQC, 2D $^1$H-$^{13}$C HSQC-TOCSY, and 2D $^1$H-$^{15}$N HSQC (see Methods section for a complete description). All the shortest peptides (iso 1a, iso 1b, iso 1c, iso 1d, iso 2a, iso 2b) as well as isoforms 1 and 3 NMR spectra were recorded at 14.1 T. Moreover, isoform 1 (25–88) and isoform 3 were also measured at 28.1 T. A high magnetic field and the use of a cryoprobe were the keys to measuring high quality spectra within one week per peptide.

The $^1$H, $^{15}$N and $^{13}$C assignments of all the peptides were performed using a combination of 2D $^1$H-$^1$H TOCSY, 2D $^1$H-$^1$H NOESY, 2D $^1$H-$^{13}$C HSQC and 2D $^1$H-$^{13}$C HSQC-TOCSY. The interpretation of the different NMR spectra allowed the quasi-complete assignment of the $^1$H, $^{15}$N, and $^{13}$C chemical shifts of the short peptides namely 75% for iso 1a, 83% for iso 1b, 89% for iso 1c, 91% for iso 1d, 83% for iso 2a and 96% for iso 2b (Table S1 in S1 File) at 14.1 T. Similarly, the assignments of 87% for isoform 3 and solely 48% for isoform 1 were performed (Table S1 in S1 File) at 28.2 T.

The $^1$H-$^{15}$N HSQC spectra of isoform 1, isoform 3 and isoform 2a and 2b are displayed in Fig 3. It is worth noticing that resonances are dispersed in a narrow spectral range of about 1 ppm, which is the hallmark of intrinsically disordered proteins (IDP) [37].

Despite the higher magnetic field (1.2 GHz), we notice that only 28 resonances are visible in the $^1$H-$^{15}$N HSQC spectrum of the isoform 1 (25–88). The same experiment was also recorded at 600 MHz without any significant change (Figure S5 in S1 File). The assigned amide resonances belong to the C-terminal residues of isoform 1, and comprise S58, T59 and the sequence L62 to T87. Note that chemical shifts of P60 and P88 were also assigned using $^1$H-$^1$H TOCSY, $^1$H-$^1$H NOESY, and $^1$H-$^{13}$C HSQC spectra. We could notice that a few other visible resonances could not be assigned due to the lack of i/i+1 correlations in the $^1$H-$^1$H NOESY spectrum. Moreover, the superimposition of short peptides $^1$H-$^{15}$N HSQC spectra shows that none of the residues from D25 to I56 (iso 1a and iso 1b) are observed in the isoform 1 spectrum whereas several residue's chemical shifts overlay nicely for isoform 1c and isoform 1d (Fig 4). Indeed, resonances of L64-Q68, Q71, K74-L77 and I85 are closely superimposed and resonances of T59, C81, and Y86 show a slight variation of their chemical shifts in the uncut isoform 1 (25–88) compared to the two peptides iso 1c and iso 1d. We also observe that the division into shorter peptides induced only a slight effect on the chemical shifts of the residues S58 to T87 and does not imply any drastic change in the electronic environment such as secondary structure change for the C-terminal part of isoform 1. Finally, we attributed the disappearance of chemical shifts of residues D25 to I56 to intermediate exchange that induces a large broadening of the chemical shifts. This intermediate exchange regime is probably due to the dynamical structural change that occurs at $k_{ex} \sim Dw$ [38].

Interestingly, residues V80, C81 and E82 display two entirely different conformations in a slow exchange regime. Indeed, we measured two independent sets of different chemical shifts (H, C, and N) for those residues (namely V80 and V80b, C81 and C81b, E82 and E82b). DMSO is a mild oxidant and cysteine residues can be in reduced or oxidized state. Effectively, we observe two sets of chemical shift values for Ca/Cb of C81, namely 55.6/26.4 ppm for the first set and 52.3/33.7 ppm for the second set (Figure S6 in S1 File). Sharma et al. demonstrated that $^{13}$C chemical shift can discriminate between the reduced and oxidized forms [39]. Indeed, from a database of 375 Ca and 337 Cb resonances it has been reported that $d_{Ca}$ (S-S) < $d_{Ca}$ (S-H) and $d_{Cb}$ (S-S) > $d_{Cb}$ (S-H). This means that C81b is in an oxidized state whereas C81 is in reduced state. Note that the intensities of the resonances for C81 are lower than for C81b,

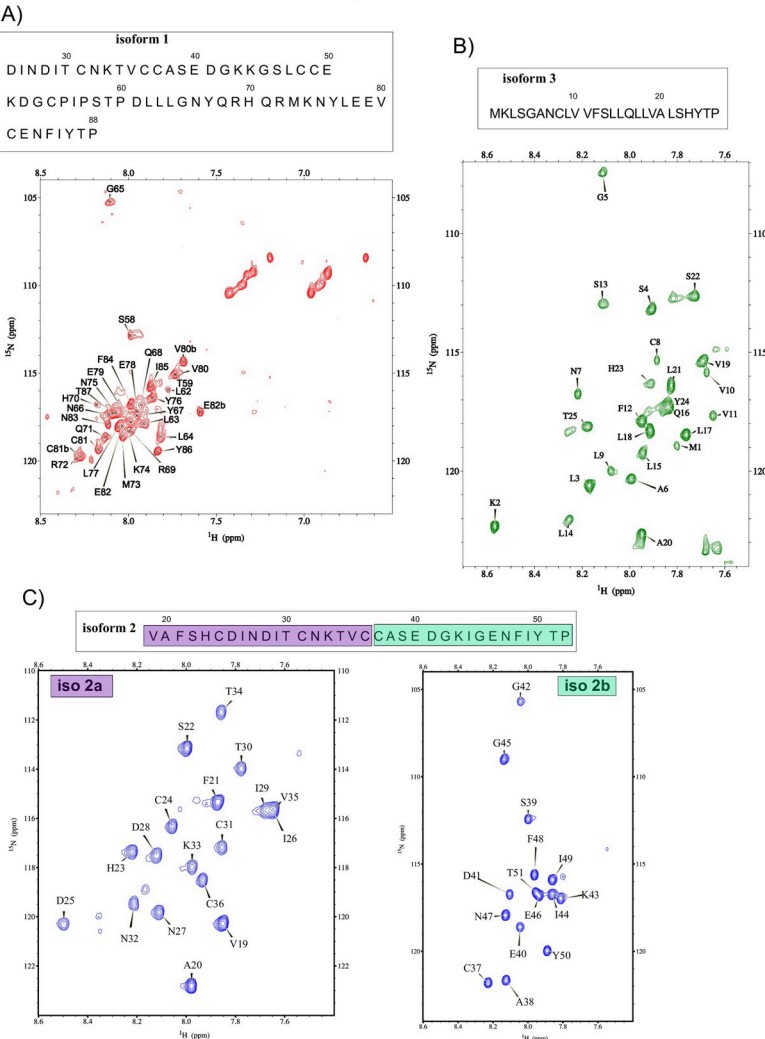

**Fig 3. 2D $^1$H-$^{15}$N HSQC spectra of MEG 2.1 isoforms.** A) isoform 1 (25–88) at a concentration of 2 mM recorded at 27˚C with a Bruker NEO spectrometer operating at a $^1$H frequency of 1.2 GHz (28.2 T). B) isoform 3 (residues 1 to 26) at a concentration of 2 mM recorded at 27˚C with a Bruker NEO spectrometer operating at a $^1$H frequency of 1.2 GHz (28.2 T). C) iso 2a (residues 19–36) and iso 2b (residues 37–52) peptides at a concentration of 2 mM recorded at 27˚C with a Varian Inova spectrometer operating at a $^1$H frequency of 600 MHz (14.1 T). The two spectrometers are equipped with a triple HCN cryoprobe. The residues numbering is displayed inside each spectrum and the pertaining sequences are displayed above each spectrum.

which indicates a major proportion of the oxidized state that is consistent with the presence of the mild oxidant solvent DMSO.

Finally, around 50% of the resonances of isoform 1 (25–88) are entirely absent in all the different spectra despite the use of two different high magnetic fields (14.1 and 28.2 T) and cryoprobe. We could ascribe the signal disappearance to the intermediate exchange regime. The use of two different magnetic fields, unfortunately, did not help to recover the missing resonances.

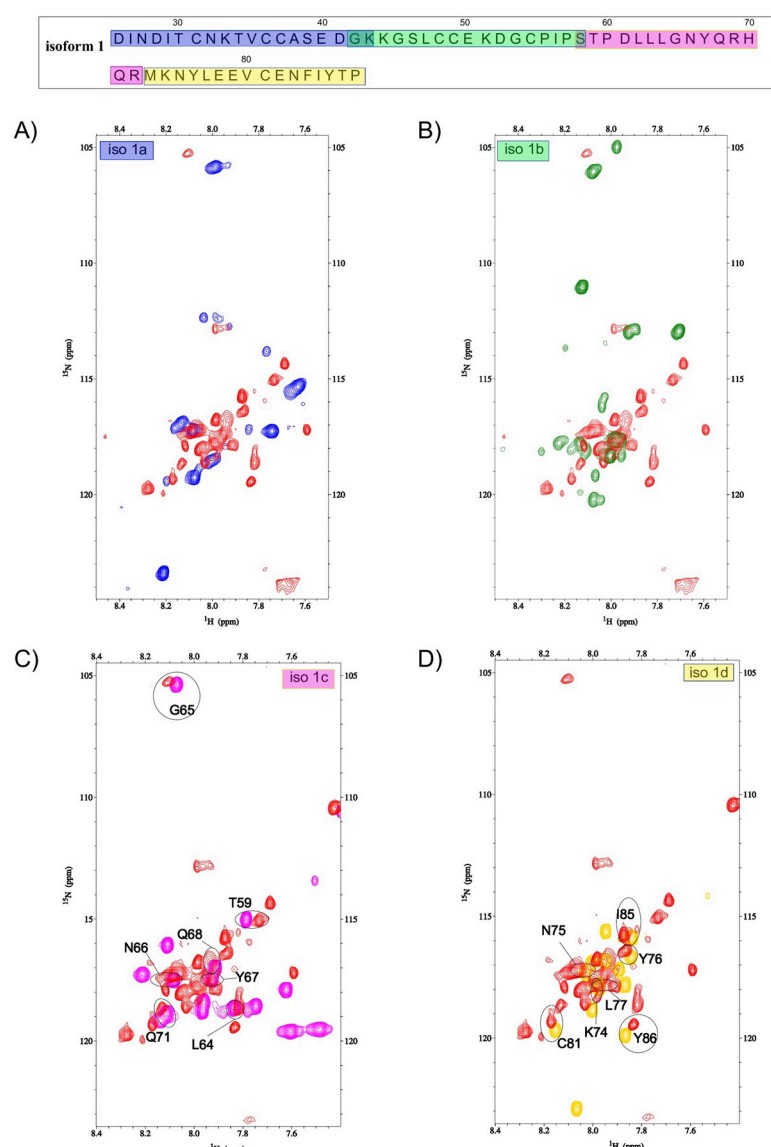

**Fig 4. $^1$H-$^{15}$N HSQC isoforms 1 overlaid spectra.** Superimposition of $^1$H-$^{15}$N HSQC spectra of isoform 1 (red) and A) iso 1a (blue), B) iso 1b (green), C) iso 1c (pink) and D) iso 1d yellow). The experiments were carried out at 28.2 T for isoform 1 and 14.1 T for the shortest peptides (iso 1a, iso 1b, iso 1c and iso 1d)). The samples were dissolved in DMSO, and the concentration of each peptide was set to 2 mM. The temperature was set to 27˚C. Residue assignments are displayed in the spectra and the isoform 1 sequence is displayed at the top of the figure. Each sequence peptide is highlighted with a different color for the sake of clarity.

## 3D NMR structures of MEG 2.1 peptides

CYANA [20] was used to derive NMR structures of iso 1a, iso 1b, iso 1c and iso 1d peptides (Fig 5A–5D) from measured NOEs (Table 2).

The Ramachandran plots indicate 0% in disallowed and generously allowed regions for all short peptides (Figure S7 in S1 File and Table 2). The highest degree of IDP is observed for iso 1a, iso 1b, and iso 1c peptides (Fig 5A–5C). Indeed, no or very few long-range distances are measured by means of NOE data analysis (Figure S8A-S8C in S1 File). Consequently, the 10-lowest energy structures don't overlay nicely, except for a short part of the sequence

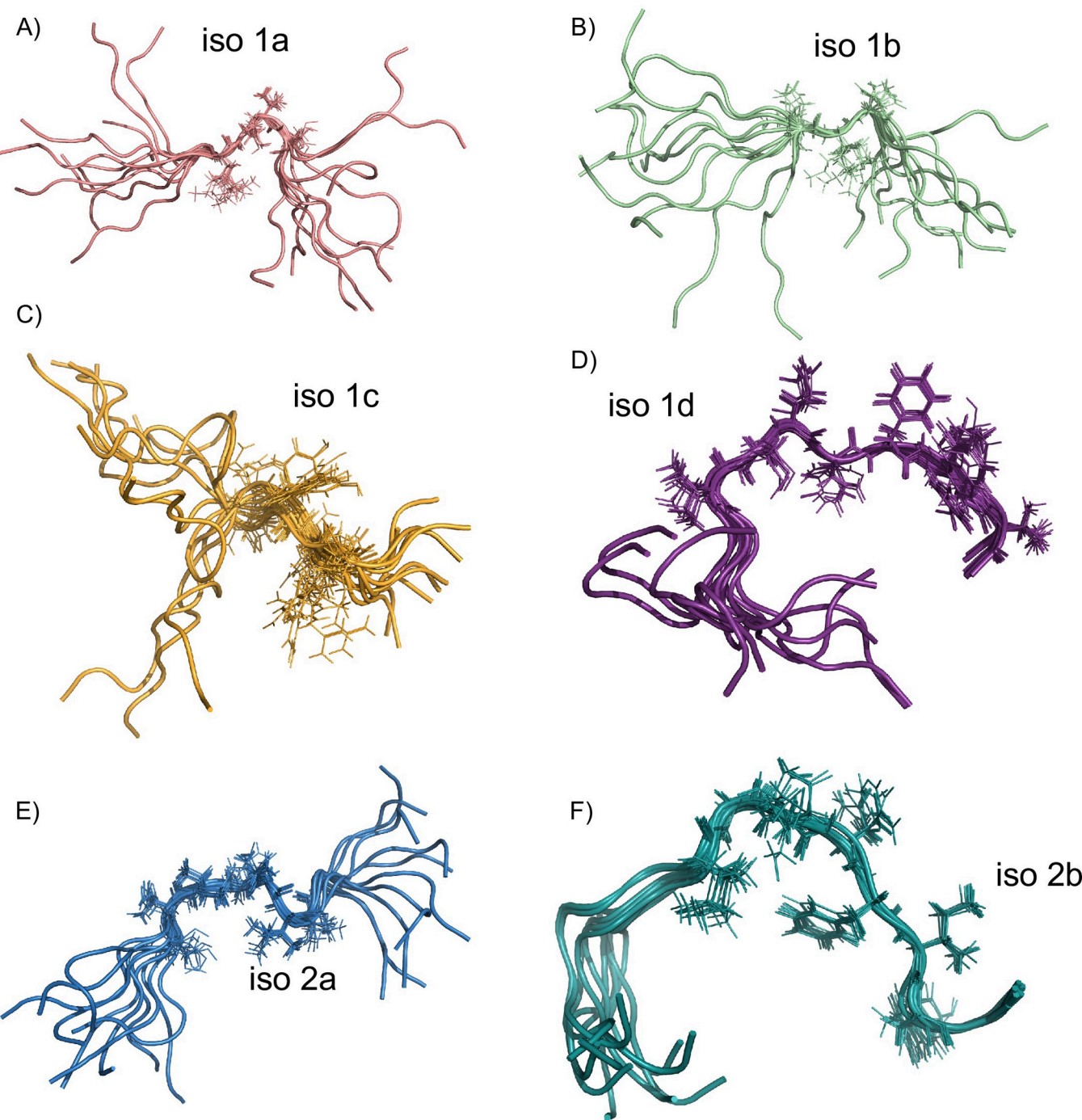

**Fig 5. NMR-derived structures of MEG 2.1 individual peptides.** The 10 lowest-energy structures derived by NMR using CYANA for A) iso 1a (aa D25-K43), B) iso 1b (aa G42-S58), C) iso 1c (aa S58-R72), D) iso 1d (M73-P88), E) iso 2a (V19-C36) and F) iso 2b (C37-P52). For each peptide, the side chains of residues exhibiting the highest number of NMR constraints are displayed in lines (residues K33-C36 for iso 1a, E50-D52 for iso 1b, N66-R69 for iso 1c, V80-T87 for iso 1d, C24-T30 for iso 2a and I44-Y50 for iso 2b).

involving barely 3–4 residues. Those peptides exhibit the lowest number of NMR constraints, uniquely i/i+1 correlations, without creating any secondary structure. On the other hand, we can notice that residues V80 to T87 of iso 1d peptide are well-stacked for the 10-lowest energy

Table 2. Total and unambiguous numbers of NOE as well as Ramachandran statistics for MEG 2.1 iso 1a, iso 1b, iso 1c, iso 1d, iso 2a and iso 2b peptides.

| Peptide | # AA | NOE numbers | | Ramachandran plots statistics | |
|---------|------|-------|-------------|-------------------|----------------------------|
| | | Total | Unambiguous | Most favored region | Additionally allowed region |
| iso 1a | 19 | 143 | 80 | 71.2% | 28.8% |
| iso 1b | 17 | 154 | 83 | 71.8% | 28.2% |
| iso 1c | 15 | 153 | 81 | 56.4% | 43.6% |
| iso 1d | 16 | 253 | 181 | 53.6% | 46.4% |
| iso 2a | 18 | 212 | 138 | 52.5% | 47.5% |
| iso 2b | 16 | 230 | 141 | 59.2% | 40.8% |

structures (Fig 5D) and we measured some few long-range distances (Figure S8D in S1 File) such as Hg-V80/ $H_N$-E78, $Hg_2$-I85/ $H_N$-T87, or $Hd_1$-I85/Ha-N83. Indeed, among 253 measured NOEs, 181 have been used by CYANA to derive the structure (Table 2). This is the highest value of NOEs used for structure calculation among the 6 peptides leading to a well-constrained structure in the middle of the iso 1d sequence.

MEG 2.1 iso 2a peptide is composed of 18 residues. Among 212 measured NOEs, the structure of the peptide was derived based on the collection of 138 unambiguous NOEs using CYANA (Table 2) [20]. The ensemble Ramachandran plot of the 10 lowest-energy structures is shown in Figure S7E in S1 File. Even in this case, there are no violated distance constraints regardless of the calculated structures, and the Ramachandran plot displays 0% in disallowed and generously allowed regions. The statistical distribution indicates 52.5% in most favored regions and 47.5% in additionally allowed regions with no phi/psi angles in the expected a-helix region (Table 2). For iso 2a peptide, i/i+1 correlations are mainly observed, with some i/i+2 correlations. A few long-range distance constraints were derived from NOEs leading to a not entirely disordered secondary structure for iso 2a peptide (residues 19–36) (Figure S8E in S1 File). Notably, we observed the highest number of constraints including short, medium, and long-range constraints for residues I26 to K33 (8–15 in Figure S8 (S1 File)). Indeed, all the 10 lowest-energy structures display a well-constrained part in the middle of the sequence involving residues C24 to T30 (Fig 5E). None of the three cysteine residues are involved in any disulfide bonds.

Iso 2b peptide consists of 16 residues (aa 37–52). In a similar manner, 230 NOEs have been measured, and 141 NOEs have been used to derive the structure of this peptide (Table 2). All those structures overlay nicely (Fig 5F). There are no violated distance constraints (see the Ramachandran plot in Figure S7F in S1 File). Here, the statistical distribution indicates 59.2% in most favored regions and 40.8% in additionally allowed regions with no phi/psi angles in the expected a-helix region as observed for iso 2a peptide (Table 2). For iso 2b peptide, we also observe mainly i/ i+1 correlations. Nevertheless, some long-distance constraints such as $Hg_{13}$-I44/$Hd_1$-Y50, $Hg_2$-I44/Ha-G42, $Hg_2$-I44/$H_N$-E40, $Hg_2$-I44/$Hd_2$-F48 or $Hg_2$-I44/$H_N$-E46 were measured. Indeed, the highest number of constraints (short, medium, and long-range) were measured for residues K43-E46, F48-I49 and T51 (7–10, 12–13 and 15 in Figure S8F (S1 File)). Consequently, the iso 2b lowest-energy structures are more constrained than the iso 2a peptide. Nevertheless, there is no secondary structure such as a-helix or b-sheet, only one turn involving residues G42, K43, I44, G45, E46, N47, and F48 (Fig 5F), is visible. Additionally, we can notice that the two NMR-derived structures of iso 2b and iso 1d are pretty much identical. This result was to be expected since the sequences of the two short peptides are identical (E82-P88 for iso 1d and E46-P52 for iso 2b). Similarly, the disordered C-terminal part of iso 1a (C37 to D41) could be compared to the disordered N-terminal part of iso 2a (C37 to D41) since the two sequences are also strictly identical due to alternative splicing.

Regarding isoform 3, the $^1$H-$^1$H NOESY spectrum (Figure S9 in S1 File) is overcrowded on the $H_N/H_a$ region since 26 $H_N$ chemical shifts gather in an extremely narrow spectral window of 1 ppm (from 7.6 to 8.6 ppm). No accurate inter-residues NOEs could be measured, despite the use of a high magnetic field (28.2 T), which prevented the calculation of the structure by CYANA.

## MD simulation of isoform 1 (25–88)

The structure of isoform 1 (25–88) was reconstructed by assembling the peptide structures of iso 1a, iso 1b, iso 1c and iso 1d with UCSF Chimera software using the default setting Join Models section for C-N peptide bond [40] (see Materials and Methods for details). This modeled structure, which is mainly disordered (Fig 6A, top), was used as the first structure for the MD simulation run. The structure was first minimized and equilibrated for 10 ns using ACEMD3 software. Then, a production run of 4 μs was performed on the system. During the first 100 ns of simulation, the structure is mainly disordered (Fig 6B) and gave rise to drastic conformational changes afterwards. Between 100 and 200 ns, we noticed the formation of short helices that encompass residues H23 to K33, S39 to L47, and E50 to G53. Interestingly, after 1 μs, those three helices merge into a unique long a-helix that encompasses residues H23 to G53 (Fig 6A, bottom, and 6B), and no conformational change occurs hereafter.

Whether or not this helix remains stable can be inferred by different parameters. First, we derived the radius of gyration (Rg) that characterizes MEG 2.1 isoform 1 from the whole 4 μs MD trajectory. As can be seen in Fig 7A, Rg first undergoes a significant compression to 1.3 nm until 1 ms prior to a marked transition that releases Rg to approx. 2 nm for the remaining simulation and indicates that the global shape of this isoform is kept constant and becomes less compact after the formation of the N-terminus helical structure. To confirm this statement, we calculated the Root Mean Square Deviation (RMSD) issued on the backbone residues either for (i) the complete isoform 1 or for (ii) the helical part encompassing residues H23 to S46 (Fig 7B). For the complete isoform 1, a smooth transition is observed that becomes clearly abrupt for the helical part at 1 μs and clearly reflects the fact that the three short helices visible until 1 ms, assemble into a unique long stable helix for the rest of the trajectory. Finally, to confirm the stability of this long a-helix, we have computed the Root Mean Square Fluctuation (RMSF) for each amino acid that captures the fluctuation about their average position starting at 1.2 ms. Fig 7C indicates that the a-helix formed by residue H23 to G53 experiences rather weak fluctuations while the rest of the protein undergoes strong fluctuations that indicate a significant flexibility over time.

This result agrees with CD data analysis since the presence of a-helical content was deduced from the CD curves. The C-terminal part of isoform 1 remains mainly disordered along the MD simulation with the presence of some small helices, bends or turns. We can notice that the highest number of NMR constraints were indeed observed for residues N66 to R69 for iso 1c and V80 to T87 for iso 1d and those two regions are part of the two small new helices L64-R69 and V80-N84 along the MD trajectory. Finally, the NMR experimental data of the last two short peptides, namely iso 1c and iso 1d, and the MD simulation converge to similar structural features for the C-terminal part of isoform 1, between residues H70 and V80 and N84-P88, where the bent elbow-like features remain constant and rigid (Figs 5C and 5D *vs* 6B and 6C).

## Discussion

Schistosomiasis is a vector-borne parasitic disease affecting over 250 million of the poorest populations in sub-Saharan Africa, Brazil, and South-East Asia and is one of the most debilitating parasitic diseases in the world with up to 700 million people at risk [1]. The disease is

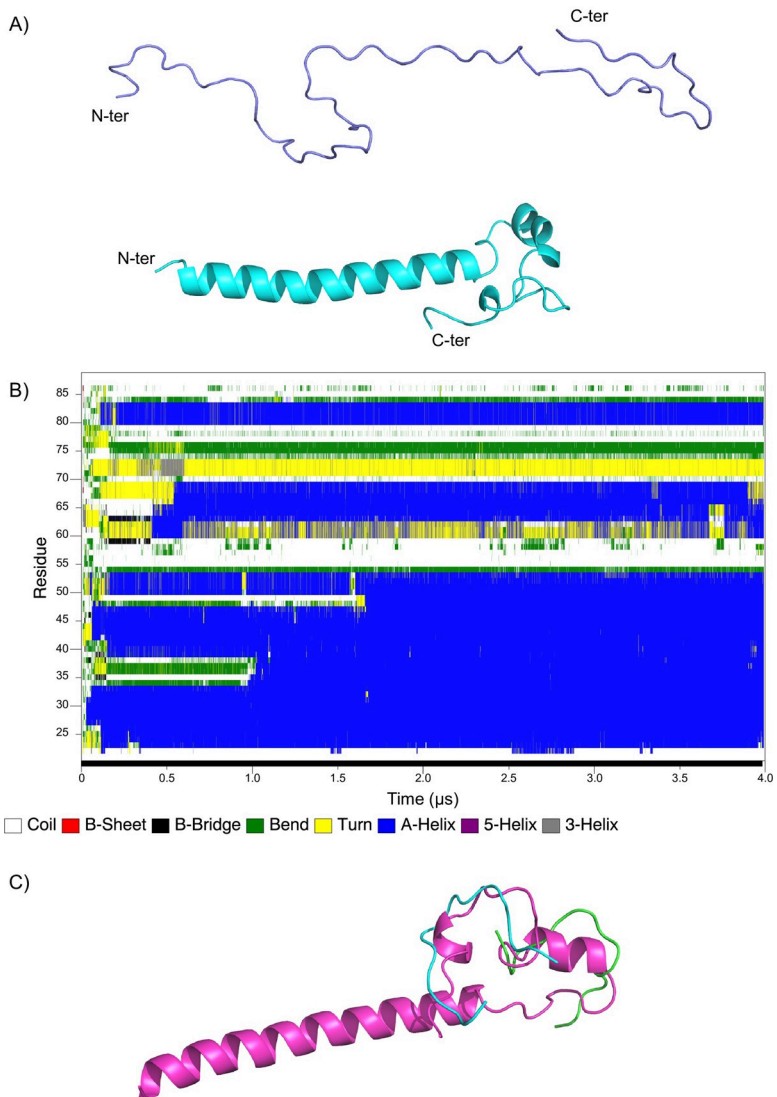

**Fig 6. Reconstructed MEG 2.1 isoform 1 MD simulation.** A) Isoform 1 reconstructed structure after minimization (top) and last frame structure after 4 μs MD simulation (bottom). B) Secondary structure evolution of isoform 1 along the MD trajectory. The secondary structure color palette is displayed at the bottom of the figure. C) Overlay of isoform 1 last frame structure (pink) with NMR-derived iso 1c (green) and iso 1d (cyan) peptides structures.

caused by trematodes with a complex lifecycle involving two hosts: a freshwater snail, in which they reproduce asexually, and a mammalian host, where sexual reproduction occurs. Each adult couple of *S. mansoni*, living in the mesenteric veins around the liver, can lay more than 350 eggs per day; these latter need to extravasate to be expelled with the feces [1, 41]. However, eggs lack motility, therefore they will secrete/excrete proinflammatory molecules, which induce the formation of clusters of immune cells, called granulomas, in the intestinal wall or liver. These formations, on one hand, assist in pushing the eggs out of the body, on the other, they lead to the chronic disease and cancer onset, if not treated [41, 42]. The analysis of secreted biomolecules highlighted, among others, the abundance of Micro-Exon Gene (MEG) proteins, a family of highly variable peptides/proteins, ranging from 7 to 20 kDa, without homologs in the host [6–8]. Of the more than 25 families encoded by the genome of *S.*

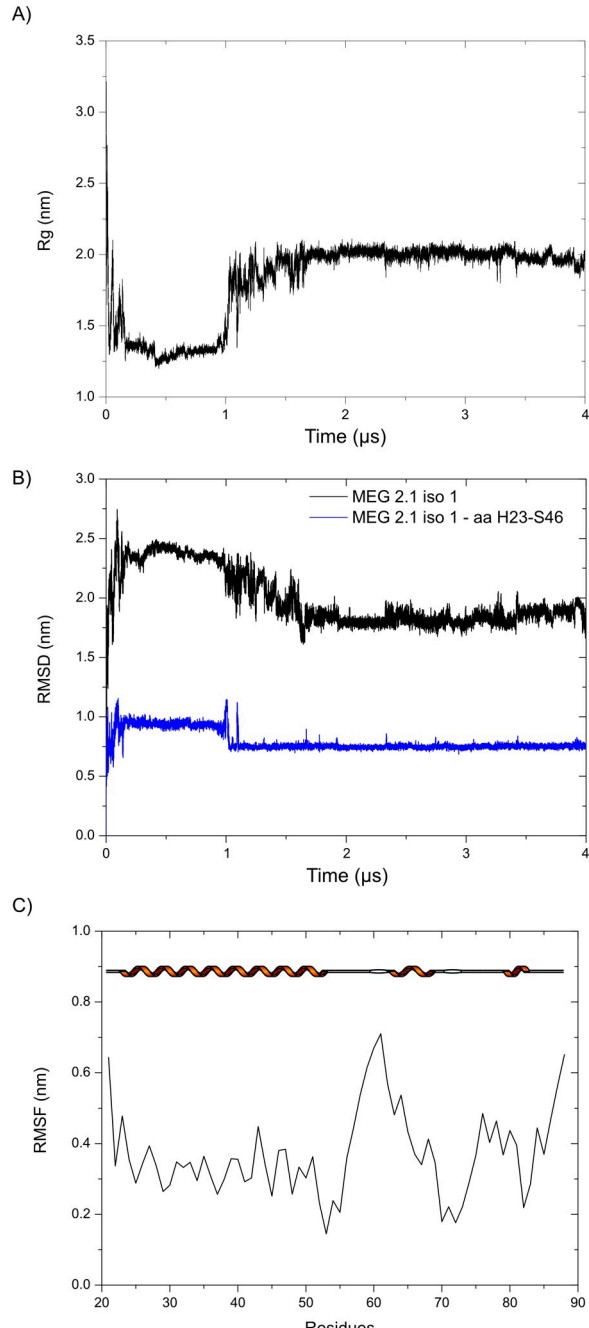

**Fig 7. Stability of the MEG 2.1 isoform 1 structure.** A) Radius of gyration (Rg) along the MD simulation trajectory of 4 μs. B) RMSD of the full MEG 2.1 isoform 1 (black) and the helical part (H23-S46, blue) along the trajectory. C) RMSF calculated from 1.2 μs, corresponding to the simulation time of the stable helix formation, to 4 μs. The cartoon representation of the secondary structure is added at the top of the graph.

*mansoni*, two are particularly abundant in the secretions, MEG 2 and MEG 3 families, each one composed of many isoforms arising from gene duplications and alternative splicing [7, 8]. It has been inferred that their high variability and their high copy number in the genome might be linked to host evasion as well as to host-parasite interactions. However, their amino acid composition makes them challenging to study *in vitro*. At present, only MEG 14, MEG 24

and MEG 27 have been biophysically characterized by CD with synchrotron light and differential scanning calorimetry, and their interaction with host factors and membranes studied [43, 44]. It is worth noticing that MEG 24 and 27 were chemically synthesized, while MEG 14, one of the longest representatives of the family, was recombinantly expressed. Also, one isoform of MEG 8 (UniProt ID Q86D79, 20 kDa) was recombinantly expressed, but used only for immunization and hemagglutination studies [45].

Sequence-wise, MEG 2.1 isoforms do not share a high identity with the four MEGs mentioned above; however, they share the IDP as well as the hydrophobic character (Fig 5 and Table 1). Given that all the tests for expression in heterologous hosts had failed, we resorted to chemically synthesize MEG 2.1 isoforms 1, 2, and 3 in order to study them *in vitro*, as was the case of MEG 24 and 27 [43]. Moreover, the synthetic isoform 1 (25–88) did not allow to get a complete assignment of the chemical shifts nor any structural information from the NOESY spectrum despite the use of a high magnetic field (28.2 T). Therefore, we had to split the sequence into four shorter peptides (iso 1a to 1d). Moreover, MEG 2.1 isoform 2 failed to be synthesized by Genosphere, hence we had to split the sequence in two, iso 2a and 2b (Fig 1). The next challenge we had to deal with the synthetic peptides was their hydrophobicity and poor solubility in biological buffers. To solve their structure by multidimensional NMR with the natural abundance of $^{15}$N and $^{13}$C, we had to solubilize all the peptides in 100% DMSO at 2 mM final concentration. Indeed, DMSO is not commonly used for NMR studies and structure resolution of peptides or proteins; it is a mild oxidant solvent known to be slightly chaotropic [46]. Nevertheless, previous papers demonstrated that the use of DMSO doesn't prevent the observation of secondary structural features nor prevents solving the 3D structure of proteins and peptides. As examples, Rath et *al.* identified unambiguously the exact localization of mycolic acid post-translational modification of *Corynebacterium* PorH using 2D $^{1}$H-$^{15}$N HSQC, 3D HSQC-TOCSY and HSQC-NOESY experiments [47]. Spyranti et *al.* solved the structure by 2D $^{1}$H NMR of myelin basic protein epitope 83–99, an immunogenic peptide involved in multiple sclerosis [48]. Takeuchi et *al.* solved the 3D structure of two cyclic Ras-binding peptides, 9A5 and 9A54, dissolved in water and DMSO, and showed that only 9A5 undergoes a conformational change [49].

In this paper, we show that MEG 2.1 derived peptides, despite being dissolved in DMSO, display peculiar 3D structural features within an overall IDP configuration. For example, iso 1d and iso 2b have elbow-like structures (Fig 5); Pro in iso 1b and 1c do not rigidify the IDP, possibly enhancing a morphing towards some a-helical content both when fused (isoform 1 from 25 to 88) and in the presence of TFE (Fig 2B). On the contrary, the two rigid elbows of iso 2a and iso 2b will prevent any geometrical rearrangement into a-helix, as seen in the CD in the presence of TFE (Fig 2B). Moreover, the 4 μs MD simulation on MEG 2.1 isoform 1 (25–88) shows a transition with a stable a-helix formation at the N-terminus and of turns/small helices at the C-terminus (Figs 6 and 7). Indeed, the superposition of the NMR spectra of the single peptides on that of isoform 1 (Fig 4) suggests a putative advantage of alternative splicing to explore diversity in 3D structure. Higher flexibility and the presence of Pro will confer morphing character to MEG IDP, especially to isoform 1, as it was the case for MEG 14 in the presence of TFE [44]. At the same time, a rigid elbow/b-turn will maintain a putative docking surface to isoform 2 and to the end of isoform 1. This diversity could play a role in host-parasite interactions. Indeed, we can notice that $^{1}$H-$^{15}$N HSQC spectra of iso 1a and iso 2a, as well as spectra of iso 1a, iso 1d and iso 2b show several chemical shift superpositions (for example $^{26}$INDI$^{29}$ and $^{84}$FIYT$^{87}$) indicating a similar structural organization (Figure S10 in S1 File).

One side mention should be made on MEG 2.1 isoform 3, which is composed of the putative signal peptide plus only 4 residues. Indeed, this isoform is helical, as expected (CD in Fig 2), however, we failed to collect sufficient accurate NMR data to solve its structure. It is also

questionable whether this isoform really exists at the protein level since it has only been found at the transcriptomic level.

## Conclusion

In this study, we present for the first time a structural demonstration that (i) chemical synthesis is a good way to circumvent problems with heterologous expression; (ii) MEG 2.1 isoforms are indeed IDP; (iii) MEG 2.1 isoform 1 (25–88) might display a chameleon behavior (morphing from IDP to a-helix) induced by the local amino acid sequence and possibly by a binding partner. Finally, there is still a long way to unveil the sequence to structure to function pathway in *S. mansoni* MEGs, in particular, and in IDPs, in general, but we have proudly set down the first stone on it.

## Supporting information

**S1 File.**
(DOCX)

## Acknowledgments

The authors thank Dr Francesca Fiorini Tregouët for her great help with the recombinant protein expression. We also thank Dr. Vojtech Vacek for the pET-22b(+) plasmid preparation and Lukas Konecny for the pET SUMO Champion plasmid preparation. We thank Kristyna Peterkova for the fruitful discussion.

## Author Contributions

**Conceptualization:** Adriana Erica Miele, Jan Dvorak, Maggy Hologne.

**Data curation:** Stepanka Nedvedova, Florence Guillière, Maggy Hologne.

**Formal analysis:** Stepanka Nedvedova, Adriana Erica Miele, François-Xavier Cantrelle, Olivier Walker, Maggy Hologne.

**Funding acquisition:** Stepanka Nedvedova.

**Investigation:** Stepanka Nedvedova, Florence Guillière.

**Resources:** François-Xavier Cantrelle.

**Software:** Olivier Walker.

**Supervision:** Adriana Erica Miele, Jan Dvorak, Maggy Hologne.

**Visualization:** Olivier Walker.

**Writing – original draft:** Adriana Erica Miele, Maggy Hologne.

**Writing – review & editing:** Stepanka Nedvedova, Florence Guillière, Adriana Erica Miele, Jan Dvorak, Olivier Walker, Maggy Hologne.

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
