## [Decision Letter · Decision Letter 0]

14 Jul 2023

PONE-D-23-18239Divide, conquer and reconstruct: how to solve the 3D structure of recalcitrant Micro-Exon Gene (MEG) protein from Schistosoma mansoniPLOS ONE

Dear Dr. HOLOGNE,

Thank you for submitting your manuscript to PLOS ONE. After careful consideration, we feel that it has merit but does not fully meet PLOS ONE’s publication criteria as it currently stands. Therefore, we invite you to submit a revised version of the manuscript that addresses the points raised during the review process.

We look forward to receiving your revised manuscript.

Kind regards,

Ravi Pratap Barnwal, Ph.D.

Academic Editor

PLOS ONE

Journal Requirements:

   "SN : Improvement in Quality of the Internal Grant Scheme at CZU, reg. no. CZ.02.2.69/0.0/0.0/19_073/0016944, financed from the funds of Operational Programme Research, Development and Education, in the framework of ESF Call no. 02_19_073 for Improving the Quality of Internal Grant Schemes at Higher Educational Institutions in priority axis 2 OP"

Reviewers' comments:

Reviewer's Responses to Questions

**Comments to the Author**

1. Is the manuscript technically sound, and do the data support the conclusions?

Reviewer #1: Yes

2. Has the statistical analysis been performed appropriately and rigorously? 

Reviewer #1: Yes

3. Have the authors made all data underlying the findings in their manuscript fully available?

Reviewer #1: Yes

4. Is the manuscript presented in an intelligible fashion and written in standard English?

Reviewer #1: Yes

5. Review Comments to the Author

Reviewer #1: The manuscript “Divide, conquer and reconstruct: how to solve the 3D structure of recalcitrant Micro-Exon Gene (MEG) protein from Schistosoma mansoni” employed a combination of biophysical techniques, such as circular dichroism and nuclear magnetic resonance (NMR) at natural abundance, along with in silico molecular dynamics simulations. By dividing the protein into shorter fragments, analyzing them individually, and reconstructing the complete structure, the authors successfully tackled the challenge of characterizing this elusive and highly variable protein class. Overall, the manuscript introduces a novel concept for investigating the structural information of MEG proteins, although some minor revision needed.

1. In the abstract, the authors mentioned that all isoforms of MEG 2.1 were analyzed using both biophysical and computational techniques. However, upon reviewing the manuscript, it is apparent that only isoform 1 was analyzed computationally. Therefore, the authors are advised to revise the sentences accordingly to accurately reflect the scope of their computational analysis.

2. The manuscript lacks some important figures, such as the root mean square deviation (RMSD), root mean square fluctuation (RMSF), and radius of gyration (Rg) graphs of the molecular dynamics (MD) trajectory. These figures are crucial for assessing the stability of the protein structure, understanding the extent of residue fluctuations, and evaluating the structural compactness of MEG 2.1. As the manuscript focuses on the structure determination of MEG 2.1 isoforms, it is essential to discuss these results and include the corresponding figures.

3. The authors should carefully review the citation of figures within the text, particularly in the MD results and discussion section. It has been observed that some citations do not match the referenced figures, causing confusion. Therefore, the authors are advised to ensure consistency and accuracy in the citation of figures throughout the manuscript.

4. The article contains grammatical and typographical errors that require scientific editing. It is recommended that the authors carefully review the manuscript to correct these errors and enhance the overall language quality.

6. PLOS authors have the option to publish the peer review history of their article (what does this mean?). If published, this will include your full peer review and any attached files.

Reviewer #1: No

While revising your submission, please upload your figure files to the Preflight Analysis and Conversion Engine (PACE) digital diagnostic tool, https://pacev2.apexcovantage.com/. PACE helps ensure that figures meet PLOS requirements. To use PACE, you must first register as a user. Registration is free. Then, login and navigate to the UPLOAD tab, where you will find detailed instructions on how to use the tool. If you encounter any issues or have any questions when using PACE, please email PLOS at figures@plos.org. Please note that Supporting Information files do not need this step.<quillbot-extension-portal></quillbot-extension-portal>

---

## [Author Response · Author response to Decision Letter 0]

18 Jul 2023

Reviewer #1

1. In the abstract, the authors mentioned that all isoforms of MEG 2.1 were analyzed using both biophysical and computational techniques. However, upon reviewing the manuscript, it is apparent that only isoform 1 was analyzed computationally. Therefore, the authors are advised to revise the sentences accordingly to accurately reflect the scope of their computational analysis.

Following the reviewers’ comment, we modified the sentence in the abstract: “We demonstrated that the combination of biophysical techniques, like circular dichroism and nuclear magnetic resonance at natural abundance, with in silico molecular dynamics simulation for isoform 1 only, was the key to solve the structure of MEG 2.1.”

2. The manuscript lacks some important figures, such as the root mean square deviation (RMSD), root mean square fluctuation (RMSF), and radius of gyration (Rg) graphs of the molecular dynamics (MD) trajectory. These figures are crucial for assessing the stability of the protein structure, understanding the extent of residue fluctuations, and evaluating the structural compactness of MEG 2.1. As the manuscript focuses on the structure determination of MEG 2.1 isoforms, it is essential to discuss these results and include the corresponding figures.

We would like to thank the reviewer to bring to our attention these missing information. Therefore, we added a new figure (Figure 7) in the main text showing the RMSD, RMSF and Rg derived from the MD trajectory. A paragraph was also added in the results part to discuss the data assessing the stability of the protein structure as well as a modified sentence in the discussion part (“Moreover, the 4 μs MD simulation on MEG 2.1 isoform 1 (25-88) shows a transition with the formation of a stable �-helix at the N-terminus and of turns/small helices at the C-terminus (Figures 6, 7)”. A sentence was also added in the Materials and Methods part concerning Rg, RMSD and RMSF calculation.

3. The authors should carefully review the citation of figures within the text, particularly in the MD results and discussion section. It has been observed that some citations do not match the referenced figures, causing confusion. Therefore, the authors are advised to ensure consistency and accuracy in the citation of figures throughout the manuscript.

Following the recommendation of the reviewer, the consistency and accuracy in the citation of the figures were checked carefully before resubmission.

4. The article contains grammatical and typographical errors that require scientific editing. It is recommended that the authors carefully review the manuscript to correct these errors and enhance the overall language quality.

To avoid grammatical and typographical errors, the manuscript was reread carefully, and we tried to do our best to polish our article.

Thus, we believe that we have addressed all reviewers’ concerns.

---

## [Decision Letter · Decision Letter 1]

19 Jul 2023

Divide, conquer and reconstruct: how to solve the 3D structure of recalcitrant Micro-Exon Gene (MEG) protein from Schistosoma mansoni

PONE-D-23-18239R1

Dear Dr. HOLOGNE,

We’re pleased to inform you that your manuscript has been judged scientifically suitable for publication and will be formally accepted for publication once it meets all outstanding technical requirements.

Kind regards,

Ravi Pratap Barnwal, Ph.D.

Academic Editor

PLOS ONE

Additional Editor Comments (optional):

Reviewers' comments:

Reviewer's Responses to Questions

**Comments to the Author**

1. If the authors have adequately addressed your comments raised in a previous round of review and you feel that this manuscript is now acceptable for publication, you may indicate that here to bypass the “Comments to the Author” section, enter your conflict of interest statement in the “Confidential to Editor” section, and submit your "Accept" recommendation.

Reviewer #1: All comments have been addressed

2. Is the manuscript technically sound, and do the data support the conclusions?

Reviewer #1: Yes

3. Has the statistical analysis been performed appropriately and rigorously? 

Reviewer #1: Yes

4. Have the authors made all data underlying the findings in their manuscript fully available?

Reviewer #1: Yes

5. Is the manuscript presented in an intelligible fashion and written in standard English?

Reviewer #1: Yes

6. Review Comments to the Author

Reviewer #1: All the queries have been addressed by the authors. The current form of the manuscript is suitable for publication.

7. PLOS authors have the option to publish the peer review history of their article (what does this mean?). If published, this will include your full peer review and any attached files.

Reviewer #1: **Yes: **Dr Prateek Pandya

<quillbot-extension-portal></quillbot-extension-portal>

---

## [Editor Report · Acceptance letter]

25 Jul 2023

PONE-D-23-18239R1 

Divide, conquer and reconstruct: how to solve the 3D structure of recalcitrant Micro-Exon Gene (MEG) protein from *Schistosoma mansoni*

Dear Dr. Hologne:

I'm pleased to inform you that your manuscript has been deemed suitable for publication in PLOS ONE. Congratulations! Your manuscript is now with our production department. 

Kind regards, 

on behalf of

Dr. Ravi Pratap Barnwal 

Academic Editor

PLOS ONE